# Quercetin Ameliorates Lipopolysaccharide-Induced Duodenal Inflammation through Modulating Autophagy, Programmed Cell Death and Intestinal Mucosal Barrier Function in Chicken Embryos

**DOI:** 10.3390/ani12243524

**Published:** 2022-12-13

**Authors:** Jinhai Yu, Guoliang Hu, Huabin Cao, Xiaoquan Guo

**Affiliations:** 1Jiangxi Provincial Key Laboratory for Animal Health, Institute of Animal Population Health, College of Animal Science and Technology, Jiangxi Agricultural University, Nanchang 330045, China; 2Department of School of Life Science, School of Life Science, Shangrao Normal University, Shangrao 334001, China

**Keywords:** Traditional Chinese Medicine, quercetin, diarrhea, autophagy, programmed cell death, intestinal mucosal barrier function

## Abstract

**Simple Summary:**

Diarrhea has been a global health problem for centuries, and its treatment has become increasingly difficult due to the antibiotics overuse and resistance. Quercetin is a common flavonoid of extracts of vegetables, fruits, and traditional Chinese herbs, however, the mechanism of quercetin alleviating LPS-induced duodenal inflammation remains elusive. The results demonstrated quercetin enhanced the inflammatory cell infiltration in the Peyer’s patch of the intestinal mucosa after LPS induction. The LPS-induced expressions of these inflammation-related factors were completely blocked by quercetin. Quercetin also decreased the protein expression of inflammatory factors after LPS induction. Quercetin could down-regulate autophagy gene expression, and decreased the protein expression after LPS induction. Quercetin treatment prevented LPS-induced increases in the gene expressions of programmed cell death factors; meanwhile, quercetin decreased the protein expression of caspase 1 and caspase 3 after LPS challenge. LPS reduced the gene expression of mucin 2, but upregulated the mRNA and protein expression of claudin 1, occludin, and ZO-1, which was balanced by quercetin. These evidences suggest quercetin can alleviate duodenal inflammation induced by LPS through modulating autophagy, programmed cell death, and intestinal barrier function.

**Abstract:**

Diarrhea has been a global health problem for centuries, and the treatment has become increasingly difficult duo to the antibiotics overuse and resistance. Quercetin is a common flavonoid of extracts of vegetables, fruits, and traditional Chinese herbs, however, the mechanism of quercetin alleviating LPS-induced duodenal inflammation remains elusive. Specific pathogen-free chicken embryos (*n* = 120) were allocated to groups including control, PBS with or without alcohol, LPS (125 ng/egg) with or without quercetin (10, 20, or 40 nmol/egg, respectively), and quercetin groups (10, 20, or 40 nmol/egg). Fifteen day-old embryonated eggs were inoculated with abovementioned solutions via the allantoic cavity. At embryonic day 19, the duodena of the embryos were collected for histopathological examination, RNA extraction and real-time polymerase chain reaction, immunohistochemical investigations, and Western blotting. The results demonstrated quercetin enhanced the inflammatory cell infiltration in the Peyer’s patch of the intestinal mucosa after LPS induction. The LPS-induced expressions of these inflammation-related factors (TLR4, IL-1β, MMP3, MMP9, NFKB1, IFNγ, IL-8, IL-6) were completely blocked by quercetin. Quercetin also decreased the protein expression of TLR4, IL-1β, MMP3, and MMP9 after LPS induction. Quercetin could down-regulate autophagy gene expression (ATG5, LC3-1, LC3-2, and LKB1), and decreased the protein expression of ATG5, and LC3-1/LC3-2 after LPS induction. Quercetin treatment prevented LPS-induced increases of the gene expressions of programmed cell death factors (TNFα, Fas, CASP1, CASP3, CASP12, Drp1, and RIPK1); meanwhile, quercetin decreased the protein expression of CASP1 and CASP3 after LPS challenge. LPS reduced the gene expression of mucin 2, but upregulated the mRNA and protein expression of claudin 1, occludin, and ZO-1, and this was balanced by quercetin. This evidence suggests that quercetin can alleviate duodenal inflammation induced by LPS through modulating autophagy, programmed cell death, intestinal barrier function.

## 1. Introduction

Diarrhea has been a global health problem for centuries. Since 1990, diarrhea has been ranked among the top ten causes of morbidity and mortality in all age groups [1]. Most diarrheal cases are caused by bacteria, parasites, or viruses, with Gram-negative bacteria, such as *Escherichia coli*, *Shigella*, *Salmonella* [2], *Campylobacter*, and *Vibrio* species among the most common pathogens. *Salmonella pullorum* is the major pathogen of chicken diarrhea that is harmful to the poultry industry in developing countries. Enterotoxigenic *Escherichia coli* is a major cause of post-weaning diarrhea in pigs and causes significant damage to the swine industry worldwide. The treatment caused by Gram-negative bacteria has become increasingly difficult due to the antibiotics overuse and resistance [3]. *Salmonella enterica* typhimurium is one of the most frequent causal pathogen of food contamination through meat animals associated to diarrhea, representing a major disease burden worldwide. The outer membrane of Gram-negative bacteria comprises lipopolysaccharide (LPS), which induces inflammation initiating macrophage activation, complement induction, and tissue factor activation. LPS could induce nucleotide oligomerization domain-like receptor pyrin domain-containing protein3 gene expression in the small intestine and large intestine of chickens [4]. The inflammation is characterized by body weight loss, liquid secretion, and inflammatory cytokine release [5]. Cytokines are the central mediators of immunological responses for the intestinal mucosa in acute and chronic intestinal inflammation. Diarrhea could impair the epithelial barrier functions, change tight junctions, promote the invasion of pathogens, induce autophagy and apoptosis of intestinal epithelia, and increase barrier permeability and exudation.

Antibiotics’ overuse and overprescription in human and veterinary medicine are common issues worldwide in the treatment and prophylaxis of bacterial infections. Residues of antibiotics in animal products may cause antibiotic resistance and allergies. Therefore, the reduction of antibiotics use in humans and livestock and poultry industries is urgently needed. In this context, it is imperative to search alternative medicines to meet this objective. There are many Chinese herbal medicines that could treat diarrhea. Traditional Chinese herbal medicines, such as Coptis Root (*Rhizoma coptidis*), danshen (*Salvia miltiorrhiza*) and their active ingredients have been shown to possess antibacterial activity in vitro. Therefore, use of alternative medicines such as phytochemicals or TCM to treat diarrhea is feasible and would reduce antibiotic use in humans, livestock, and poultry industries. 

Network pharmacology is a relatively new discipline based on network biology, system biology, polypharmacology and bioinformatics, and provides a new tool for drug discovery, development and research. It integrates a range of data and subjects and provides analyses of complex relationships and interactions of multiple drug targets, signaling pathways, etc. Network pharmacology is a powerful tool for studying TCM, which takes a holistic approach of disease treatment. Based on the concept of network pharmacology, the Traditional Chinese Medicine Systems Pharmacology Database and Analysis Platform (TCMSP) was created to promote and facilitate research of traditional Chinese medicines and drug discovery. TCMSP, together with other online databases, such as SwissTargetPrediction, OMIM, and GeneCards, as well as the Database for Annotation, Visualization and Integrated Discovery (DAVID), are powerful tools for predicting and analyzing drug targets and signaling pathways. 

In this study, we selected Chinese herbal medicines with antidiarrheal effects from the TCM literature, identified active ingredients of the herbal medicines using the TCMSP database, then, gene targets of the active ingredients were predicted on SwissTargetPrediction, and genes associated with diarrhea were obtained on OMIM (https://omim.org/ (26 January 2021) and GeneCards (https://www.genecards.org/ (accessed on 6 February 2021). Genes predicted from the above databases were analyzed using the Cytoscape software(version 3.7.2), and gene targets of the ingredients matching the genes associated with diarrhea were subjected to gene ontology (GO) and Kyoto Encyclopedia of Genes and Genomes (KEGG) pathway enrichment analyses in the Database for Annotation, Visualization and Integrated Discovery (DAVID) to predict molecular biology functions, biology process, and cytology component, and signaling pathways. 

## 2. Materials and Methods

### 2.1. Chinese Herbal Medicines with Antidiarrheal Effects and Key Active Ingredients

Chinese herbal medicines with antidiarrheal effects were identified by searching the Compendium of Materia Medica, Traditional Chinese Medicines and Chinese Clinical Medicine Dictionary using the keywords “dysentery” or “diarrhea”. A total of 28 Chinese herbal medicines with anti-diarrhea effects were identified by the literature search (Table 1).

The selected Chinese herbal medicines were analyzed for bioactive ingredients in the Traditional Chinese Medicine Systems Pharmacology (TCMSP; https://tcmspw.com/tcmsp.php (accessed from 10 November 2019 to 22 February 2021) database using the following screening criteria: (1) the oral bioavailability should be 30% or higher, and (2) drug-likeness should be greater than or equal to 0.18. Each selected Chinese herbal medicine contains 2 to 92 bioactive ingredients, yielding a total of 291 bioactive ingredients from the 28 Chinese herbal medicines. The 291 active ingredients were analyzed on Cytoscape (version 3.7.2). Three active ingredients, quercetin, kaempferol, and beta-sitosterol, were selected based on the highest networking node degrees. 

### 2.2. Analysis of Potential Targets and Signaling Pathways

Potential target genes of the identified ingredients, quercetin, kaempferol and beta-sitosterol, were predicted on SwissTargetPrediction (http://www.swisstargetprediction.ch/ (accessed on 2 July 2020)). Potential genes associated with diarrhea, dysentery or bacterial dysentery were obtained from OMIM (https://omim.org/ (accessed on 26 Januray 2021) and GeneCard (https://www.genecards.org/ (accessed on 6 February 2021). The potential pharmacological targets of the 3 ingredients and the genes associated with diarrhea were analyzed using the Cytoscape software (3.7.2) for common genes from the 3 databases. 

The target genes were subjected to gene ontology (GO) and Kyoto Encyclopedia of Genes and Genomes (KEGG) pathway enrichment analyses in the Database for Annotation, Visualization and Integrated Discovery (DAVID) to predict molecular biology functions, biology processes, cytology components, and signaling pathways (https://david-d.ncifcrf.gov/tools.jsp (accessed on 22 September 2020).

### 2.3. Reagents, Chicken Embryos and Experimental Design

We found quercetin, kaempferol, and beta-sitosterol in 28 Chinese herbal medicines with antidiarrheal activity. The biological processes of diarrhea were involved in inflammatory responses. The TNF signaling pathway plays an important pole in diarrhea, which contains apoptosis, autophagy, necroptosis and inflammatory cytokines’ expression. To verify in silico analysis findings, LPS from *Salmonella enterica* serotype typhimurium was selected to induce duodenal inflammation in embryonated chicken eggs to simulate diarrhea, and quercetin, as a representative active ingredient, was administered to demonstrate antidiarrheal effects via modulating autophagy, apoptosis, necroptosis, and intestinal mucosal barrier functions.

LPS from *Salmonella enterica* serotype typhimurium (*S. typhimurium*, product number: L7261, Sigma-Aldrich Trading Co., Ltd., Shanghai, China) was dissolved in phosphate-buffered solution (PBS) at 0.6 μg/mL (125 ng/egg). Quercetin (Product number: Q4591, Sigma-Aldrich Trading Co., Ltd., Shanghai, China) was dissolved in 100% ethanol at 50, 100 and 200 μM (10, 20 and 40 nmol/egg). 

Because the chick genome demonstrates remarkable evolutionary conservation with mammals, the expression patterns of several genes and proteins are well-conserved between chick and mouse embryos. In addition, injection into the allantoic cavity of chicken embryos was an ideal method to avoid the interaction of environmental LPS and intestinal LPS from gut microorganisms. Therefore, the chicken embryos were selected for the present study. Specific pathogen-free babcock embryos (weight 56.76 ± 3.32 g) were provided by a chicken breeder (Ji’nan SAIS Poultry Co., Ltd., Ji’nan, Shandong, China). The antibodies to many viruses and bacteria were negative in these embryos, including adenovirus group I and group III, avian influenza type A, reovirus, anemia virus, fowl pox, infectious bronchitis virus, infectious bursal disease, infectious laryngotracheitis virus, lymphoid leucosis virus A, B, and J, Marek’s disease (serotype 1, 2, 3), *Mycoplasma gallisepticum*, *Mycoplasma synoviae*, Newcastle disease virus, reticuloendotheliosis virus, and *Salmonella pullorum gallinarium*. The fertilized eggs were individually weighed and divided into 10 groups, each group consisting of 4 replicates with 3 eggs per replicate. The eggs were incubated under standard conditions (temperature: 38 °C, humidity: 60–70%). All eggs were candled and weighed at embryonic day 7 and embryonic day 14 to eliminate undeveloped eggs. They were untreated or injected with 0.2 mL/egg of PBS, LPS (125 ng/egg; 0.2 mL/egg), PBS and ethanol (0.2 mL each per egg), quercetin + LPS (10, 20 and 40 nmol + LPS 125 ng/egg), and quercetin (10, 20 and 40 nmol/egg). Each treatment was administered to 15 day-old embryonated eggs by injection into the allantoic cavity according to the procedure described by a previous study [6]. The injection was conducted in a vertical clean bench after disinfection with 75% alcohol and 1% povidone iodine solution in 75% alcohol. The injection hole was sealed by paraffin before the eggs were returned to the incubator. All eggs were injected by the same individual to reduce experimental variation. The duration of eggs outside the incubator for weighing, examination, and injection of the treatment solution was approximately 10 min. 

At embryonic day 19, the duodenum of the embryos was collected for histological examination and RNA extraction was performed for real-time quantitative polymerase chain reaction (qPCR). The duodenum tissues for histological examination were processed by a routine method (see below). The sample for PCR was stored in liquid nitrogen until RNA extraction. The study was approved by the Jiangxi agricultural University Animal experiment Ethics Committee (JXAULL-2022002).

### 2.4. Histology

Gut tissues were fixed in 4% paraformaldehyde, dehydrated, embedded in paraffin blocks, sectioned to 3 μm-thick sections (model:RM2016, Shanghai Leica Instrumental Ltd., Shanghai, China), mounted on slides, and stained with haematoxylin and eosin following established histology procedures. The slides were scanned by a Pannoramic DESK (3D HISTECH Ltd., Empty Coolidge Ave, Budapest, Hungary) with the panoramic scanner software.

### 2.5. qPCR

Total RNA was extracted from liquid nitrogen-frozen duodenum (50 mg) using the TransZol Up Plus RNA kit (Catalog-number: ER501-01, TransGen Biotech Co., Ltd., Beijing, China). Absorbance at 230, 260 and 280 nm was measured by spectrophotometry (NanoDrop2000, Thermo Fisher Scientific, Waltham, MA, USA) for the assessment of RNA purity. The extract with both OD260/280 nm (2.07 ± 0.03) and OD260/230 nm ratios (2.20 ± 0.12) was acceptable for PCR analysis. First-strand cDNA was synthesized from total RNA (800 ng) with an EasyScript^®^One-step gDNA removal and cDNA Synthesis SuperMix kit (Catalog-number: AE311-03, TransGen Biotech, Beijing, China) by a T100 thermal cycler (BIO-RAD Laboratories, Inc., CA, USA) according to the manufacturer’s protocol. The mRNA levels of genes were determined by real-time quantitative PCR using a BioRad CFX384 Real-Time system (Model No.: CFX384^TM^ Optics Module, BIO-RAD Laboratories, Inc., CA, USA) or BioRad CFX Connect Real-Time system (Model No.: Connect^TM^ Optics Module, BIO-RAD Laboratories, Inc., CA, USA). A total of 33 genes were selected to study the gene intestinal inflammatory factors, autophagy, apoptosis, necroptosis, and intestinal mucosal barrier function. The sequence of genes was obtained from the USA National Center for Biotechnology Information web (NCBI, https://www.ncbi.nlm.nih.gov /nuccore/ (accessed on from 5 June 2021 to 15 September 2021), and the forward and reverse primers were obtained by Primer-BLAST (https://www.ncbi.nlm.nih.gov/tools/primer-blast/ (accessed on 5 June 2021 to 15 September 2021). The primers are listed in Table 2. For real-time quantitative PCR, 2 μL of isolated template was added to the PCR reaction mixture, which contains 10 μL 2× PerfectStart^®^Green qPCR SuperMix (Catalog-number: AQ601-02, TransGen Biotech, Beijing, China) and 0.2 µM of each primer (0.4 μL /primer). PCR reactions consist of 1 cycle at 94 °C for 30 s and 43 cycles at 94 °C for 5s and an annealing temperature of 60 °C for 15 s and 72 °C for 10 s. Glyceraldehyde-3-phosphate dehydrogenase (GAPDH) was used as the housekeeping gene. The relative levels of target mRNA expression were calculated using the 2^−∆∆Ct^ method.

### 2.6. Immunohistochemistry Investigation

Immunohistochemical investigations were carried out using an indirect method of peroxidase with a primary antibody specific for CASP1 (caspase 1, anti-CASP1, GB11383, Servicebio, Wuhan servicebio technology Co., Ltd., Wuhan, Hubei province, China), CASP3 (caspase 3, anti-CASP3, GB11532, Servicebio, the detailed information of Servicebio company was the same as abovementioned one), Claudin 1 (anti- Claudin 1, GB113685, Servicebio), LC3-1/LC3-2 (microtubule associated protein 1 light chain 3 alpha/beta, anti- LC3-1/LC3-2, GB11124, Servicebio), MMP9 (matrix metallopeptidase 9, anti- MMP9, GB11132, Servicebio), and TLR4 (toll like receptor 4, anti- TLR4, GB11519, Servicebio).

Paraffin sections were deparaffinized and rehydrated, and sections were kept in epitope retrieval solution (pH = 6.0 citric acid), and heated and boiled with moderate power by a microwave oven for 8 min, cooled for 8 min, and heated with low power for 7 min, then were washed in PBS (pH = 7.4) in a decoloration shaker for 3 times (5 min per time). Serial sections were incubated with 3% H_2_O_2_ in room temperature for 25 min in dark place, then were washed in PBS (pH = 7.4) in a decolorizing shaker for 3 times (5 min per time) to eliminate endogenous peroxidase activity.

They were then rinsed and blocked by 3% normal bovine serum albumin (G5001,Servicebio; Wuhan servicebio technology Co., Ltd., Wuhan, Hubei province, China) in room temperature for 30 min. Sections were rinsed and incubated with a primary antibodies, anti-CASP1 (1:1000), anti-CASP3 (1:500), anti-Claudin 1 (1:500), anti-LC3-1/LC3-2 (1:200), anti-MMP9 (1:500), and anti- TLR4 (1:1000), in a moist chamber at 4 °C for 12 h. Sections were washed in PBS (pH = 7.4) in a decoloration shaker for 3 times (5 min per time) and centrifuged. Following this, the sections were rinsed and incubated for 50 min at room temperature in goat and anti-rabbit immunoglobulins (IgG) that were horseradish peroxidase (HRP)-conjugated (1:200; GB23303, Servicebio). Sections were washed in PBS (pH = 7.4) in a decolorizing shaker for 3 times (5 min per time). The slides were incubated with diaminobenzidine solution (DAB, G1211, Servicebio) and we terminated the stain by running water. Then, they were counterstained by hematoxylin solution for 3 min and rinsed by running water. After rinsing in PBS, slides were dehydrated, mounted, and examined under a microscope (E100, Nikon, Tokyo, Japan) equipped with a Nikon Digital imaging system (Nikon DS-U3, Nikon, Tokyo, Japan).

### 2.7. Western Blot Analysis

Fresh tissues were homogenized, washed in phosphate-buffered saline (PBS) solution and lysed in RIPA lysis buffer (G2002, Servicebio) supplemented with protease inhibitor (G2007, Servicebio) (RIPA:phenyl methyl sulfonyl fluoride: protein phosphatase inhibitor = 100:1:1). The homogenates were lysed on ice for 30 min, and vibrated once per 5 min, centrifuged at 14,170× *g* at 4 °C for 10 min, and we collected the supernatant. The total protein concentration of the dissolved lysate was quantified using the bicinchoninic acid protein assay kit (G2026, Servicebio). Protein loading buffer was added (total protein: loading buffer = 4:1) and heated for 15 min at 100 °C, then cooled to room temperature, and stored in a −20 °C refrigerator. A sample of 20 μg protein was electrophoresed using polyacrylamide gel (PAGE) and transferred onto a polyvinylidene difluoride (PVDF) membrane using a wet transfer electrophoresis apparatus (model: SVT-2, Servicebio). The blots were then blocked with 5% skimmed milk PBST for 30 min followed by overnight binding with specific antibodies at 4 °C, incubation with the secondary antibody for 30 min, and extensive washing. The specific antibodies were used to detect TLR4 (1:3000, Servicebio), IL-1β (1:3000, BIOSS), MMP9 (1:3000, Servicebio), MMP3 (matrix metallopeptidase 3, 1:3000, Servicebio), CASP1 (1:3000, Servicebio), CASP3 (1:3000, Servicebio), claudin 1 (1:3000, Servicebio), occludin (1:3000, Servicebio), β-actin (1:5000, Servicebio), and the secondary IgG HRP-conjugated antibody (1:5000; Servicebio). The expression levels of the target proteins were normalized to β-actin.

### 2.8. Statistical Analysis

All data were statistically analyzed using the *SPSS* software (version 16.0, SPSS Inc., Chicago, IL, USA). The data are presented as mean ± standard deviation, and were analyzed using the paired T test method. Differences at *p* < 0.01 were considered significant. 

## 3. Results

### 3.1. Bioactive Ingredients and Targets of Chinese Herbal Medicines

A total of 291 bioactive ingredients were found for 28 Chinese herbal medicines by TCMSP. Using Cytoscape interaction network analysis, the top three ingredients with the highest node degrees were quercetin (MOL000098, node degree = 23), kaempferol (MOL000422, node degree = 19) and beta-sitosterol (MOL000358, node degree = 19) (Figure 1), which were selected for further analysis. The node degree of other ingredients was lower than 19.

We analyzed the potential biological targets of quercetin, kaempferol and beta-sitosterol on SwissTargetPrediction (http://www.swisstargetprediction.ch/ (accessed from 2 July 2020). There were 99, 98, and 15 potential targets predicted for quercetin, kaempferol and beta-sitosterol, respectively. We also obtained genes for diarrhea, dysentery or bacterial dysentery on OMIM (https://www.omim.org/ (accessed on 26 January 2021) and GeneCards (https://www.genecards.org/ (accessed on 6 February 2021). The number of genes for diarrhea, dysentery or bacterial dysentery was 797 by OMIM, and 3253 by GeneCards. The potential biological targets of the three ingredients and genes associated with diarrhea were analyzed by Cytoscape network analysis to screen for common genes from all three databases. A total of 15 mutual target genes with a network node degree greater than four were identified (Table 3). 

GO and KEGG enrichment analysis revealed that the hub target or genes of biological processes included inflammatory response, astrocyte activation, positive regulation of protein phosphorylation, cyclin-dependent protein serine/threonine kinase activity and protein autophosphorylation (score ≥ 2). The molecular functions involved RNA polymerase II transcription factor activity, enzyme binding, nitric-oxide synthase regulator activity, receptor signaling protein tyrosine kinase activity and protein binding of molecular function. The estrogen signaling pathway and tumor necrosis factor (TNF) signaling pathway were the main pathways (Figure 2).

### 3.2. Effects of Quercetin on Intestinal Mucosa after LPS Induction in Chicken Embryos

When the LPS dose was 125ng, there was bleeding in lamina propria of intestinal epithelial cells; there was inflammatory cell infiltration in the Peyer’s patch in LPS group. In quercetin group, intestinal epithelial cells had no inflammatory cells infiltration. (Figure 3).

### 3.3. Quercetin Ameliorates Gut Inflammation through Modulating Autophagy, Programmed Cell Depth, Barrier Function in Chicken Embryos

As expected, LPS induced pattern recognition receptor TLR4 and nuclear factor kappa B subunit 1(NFKB1) mRNA expression, as well as the mRNA expression of inflammatory cytokine interferon gamma (IFNγ) and the chemoattractants interleukin-8 (IL-8), interleukin-6 (IL-6), and interleukin-1β (IL-1β), MMP3, and MMP9. The LPS-induced expression of these inflammation-related factors was completely blocked by quercetin at all doses. Quercetin or LPS had no effects on myeloid differentiation primary response 88 (MYD88), an inflammatory cytokine, and quercetin did not alter LPS-induced down-regulation of interleukin-10 (IL-10), an anti-inflammatory cytokine (Figure 4). The immunopositivity of TLR4, MMP3, and MMP9 in the villus, crypt, lamina propria, Peyer’s patches, tunica muscularis, and serosa of intestinal mucosa increased after LPS induction compared with that of control group, whereas the immunopositivity to TLR4, MMP3, and MMP9 in the treatment group significantly decreased when compared with that of the LPS group (Figure 5, Figure 6 and Figure 7). The protein expression of TLR4, IL-1β, MMP3, and MMP9 increased after LPS challenge compared with that of control group, whereas quercetin could decrease these expressions when compared with the LPS group (Figure 8). These findings clearly demonstrated the anti-inflammatory activity of quercetin in our intestinal inflammation model.

Autophagy activity was significantly increased by LPS, as indicated by markedly increased mRNA expression of autophagy related gene 5 (ATG5), LC3-1, LC3-2, and serine/threonine kinase 11 (LKB1) (Figure 9). The immunopositivity of ATG5, and LC3-1/ LC3-2 in the villus, crypt, lamina propria, Peyer’s patches, tunica muscularis, and serosa of the intestinal mucosa increased after LPS induction compared with that of control group, whereas the immunopositivity to ATG5, and LC3-1/ LC3-2 in treatment group significantly decreased when compared with that of the LPS group (Figure 10 and Figure 11).

Consistent with the inflammation of duodenum, the death receptors Fas cell surface death receptor (Fas) and ligand (TNFα), B cell CLL/lymphoma 2 (Bcl-2), as well as the caspases (CASP1, CASP3, CASP12), cell death regulator dynamin 1 like (Drp1), mitochondrial fission protein, and receptor interacting serine/threonine kinase 1 (RIPK1), involved in apoptosis, pyroptosis and necroptosis, were upregulated by LPS. Quercetin treatment significantly prevented LPS-induced increases of these programmed cell death factors (Figure 12). The down-regulation of TNFα by quercetin was dose-dependent, while no dose relationship was present for CASP1 and CASP3. The upregulation of the necroptosis factors RIPK1 and Drp1 by LPS was completely blocked by quercetin at all test doses; the inhibitory activity of quercetin against these factors was not dose-dependent. The immunopositivity of CASP1 and CASP3 in the villi, crypts, lamina propria, Peyer’s patches, tunica muscularis, and serosa of the intestinal mucosa increased after LPS induction compared with that of control group, whereas the immunopositivity to CASP1 and CASP3 in the treatment group significantly decreased when compared with that of LPS group (Figure 13 and Figure 14). The protein expression of CASP1 and CASP3 increased after LPS challenge compared with that of control group, whereas quercetin decreased these expressions compared with the LPS group (Figure 15).

LPS reduced the expression of mucin 2, a major mucus component of the intestinal villus, but considerably upregulated the epithelial tight junction tight proteins, zonula occludens-1 (ZO-1), claudin 1 and occludin (Figure 16). Quercetin, when administered without LPS, also down-regulated mucin 2, but it significantly counteracted the effect of LPS on mucin 2 expression. Furthermore, quercetin at the low dose (10 nmol/egg), when administered with LPS, increased mucin 2 expression by 8.5-fold compared with the sham and vehicle control groups, while the high dose of quercetin (40 nmol/egg) had a much lesser effect on LPS-induced reduction of mucin 2. The immunopositivity of claudin 1 and ZO-1 in the villus, crypt, lamina propria, Peyer’s patches, tunica muscularis, and serosa of intestinal mucosa increased after LPS induction compared with that of the control group, whereas the immunopositivity to claudin 1 and ZO-1 in treatment group significantly decreased which compared with that of LPS group (Figure 17 and Figure 18). The protein expression of claudin 1 and occludin increased after LPS challenge compared with that of the control group, whereas quercetin could decrease these expressions compared with the LPS group (Figure 19).

## 4. Discussion

Quercetin is a common flavonoid of extracts of vegetables, fruits, and traditional Chinese herbs; it has antioxidant, anti-inflammatory, hepatoprotective, and anti-cancer effects [8]. It is reported that quercetin dietary supplementation has an antioxidation effect on meat of broiler chicken [9]. Another study indicated quercetin supplementation altered the elemental profiles in laying hens [10]. However, quercetin’s antidiarrheal effect had not been reported. We found quercetin, kaempferol, and beta-sitosterol in 28 Chinese herbs had antidiarrheal effects through network pharmacology. Evidence indicated quercetin and kaempferol had bacteriostatic effects on Gram-positive and Gram-negative bacteria, such as *Streptococcus mutans* [11], *Salmonella enterica* serotype typhimurium, and *Escherichia coli* [12]. In the present study, the quercetin was selected to elucidate the molecular mechanism of the antidiarrheal effect, and we need further study to testify another two bioactive compounds in future. We used the LPS from *Salmonella enterica* serotype typhimurium to simulate the intestinal inflammation, and revealed quercetin attenuated the intestinal inflammation. Zhang et al. found quercetin derived from the methanol extract of hawthorn presented antibacterial activity against *Staphylococcus aureus* [13]. We found that quercetin is also rich in hawthorn leaf with antidiarrheal effects. It indicated the quercetin could be extracted from hawthorn fruit and leaves with antibacterial or antidiarrheal effects. Previous report indicated dingxiang (clove, *Syzygium aromaticum*) had an antibacterial effect on multidrug-resistant bacteria, including *Klebsiella pneumoniae* and most *Staphylococcus aureus* strains [14]. Our study revealed quercetin was one of the most powerful ingredients in dingxiang with antidiarrheal effects. Meanwhile, the antidiarrheal mechanism of quercetin was analyzed in this study. We found the biological processes of diarrhea were involved in inflammatory responses from GO and KEGG analysis. It indicates the diarrhea is involved in intestinal inflammation. Quercetin is one of the main bioactive ingredients in 28 Chinese medicinal herbs, which are commonly used in the treatment of diarrhea. Diarrhea is a clinical symptom of gastrointestinal inflammation from a variety of causes, commonly bacterial infections. LPS is one of the main constituents of Gram-negative bacterial membrane and induces intestinal inflammation. In our intestinal inflammation model, we demonstrated that quercetin inhibited LPS-induced duodenal inflammation via multiple mechanisms. 

The intestinal mucosal barrier function is the first line of defense against pathogens, nutritional deficiency, and stress. The elements of mucosal barrier function have four parts: epithelia, mucus layer, mucosal immune system, and microbiota. Mucosal barrier function depends largely on dynamic balance of epithelia cell proliferation, differentiation, migration, apoptosis, and shedding. The renewal process of intestinal epithelia lasts approximately one week. Chicken embryos start to ingest the amniotic fluid around 17 days of incubation, stimulating the rapid development of intestinal mucosa [15]. In our intestinal inflammation model, we injected LPS and/or quercetin into the allantoic cavity at 15 days of incubation just before the ingestion of amniotic fluid. LPS induced duodenal mucosa inflammation structurally characterized by inflammatory cell infiltration into the Peyer’s patches in the LPS group. Quercetin enhanced the inflammatory response compared to the untreated control group. This suggests that quercetin may improve intestinal health and could be used as a digestive aid or alternative medicine to treat diarrhea.

The mucosal immune system consists of various types of leukocytes and dendritic cells. The recruitment and activation of leukocytes and dendritic cells will promote intestinal inflammation. Pattern recognition receptors such as TLR and nucleotide-binding oligomerization domain proteins (NOD) play a pivotal role in mediating host immune responses. Cumulative evidence has indicated that LPS could activate the TLR4/NFKB signaling pathway. TLR4 and NFKB gene expression increased in the intestine of chicks after *Salmonella enterica* serovar pullorum infection [16] and *Salmonella enterica* serotype typhimurium infection [17]. Our study found the TLR4 mRNA and protein expression increased after LPS induction, and quercetin could improve and restore the immune response. Cytokines are the central mediators of immunological responses in the intestinal mucosa in acute and chronic intestinal inflammation [18]. IL-1β signals through various adapter proteins and kinases that lead to activation of downstream target NFKB. In the present study, the mRNA and protein expression of IL-1β increased after LPS challenge, which promoted NFKB expression to increase. Quercetin reduced the NFKB mRNA expression increase after LPS induction, similar to the results of a previous report [19]. LPS induces neutrophils, monocytes, and epithelial cells to release pro-inflammatory cytokines, including IL-8. IL-8 peaked at the onset of disease in stool of patients with severe acute shigellosis [20], which induced LPS production initiating IL-8 expression. Consistent with evidence of duodenal inflammation, the mRNA expression of inflammatory cytokines (TNFα, IL-1β, IL-6, IL-8, and IFNγ) was upregulated by LPS. LPS as a ligand of TLR4 activates the TLR4/NFKB signaling pathway [21]. As expected, LPS induced the expression of both TLR4 and NFKB1, as well as p38 MAP kinase, which regulates the synthesis of several important pro-inflammatory factors in chicken embryos in our study. Quercetin inhibited the upregulation of the inflammatory cytokines as well as TLR4, NFKB1 and p38 MAPK by LPS probably through the blockade of the binding of LPS to TLR4. However, this hypothesis requires verification by further studies.

In addition, IL-10 production, derived from macrophages, dendritic cells, and T cells, increased after LPS challenge. IL-10 knock-out mice spontaneously develop intestinal inflammation [22]. Accordingly, IL-10, an immunomodulatory cytokine, plays a crucial pole in maintaining intestinal homeostasis. Porcine ileum mucosa infected with *Salmonella typhimurium* revealed a massive inflammatory response: the IL-1β and IL-8 were overexpressed [23], similar to our results. The mRNA expression of IL-10, an anti-inflammatory cytokine, was down-regulated by both LPS and quercetin. IL-10 expression is closely related to TNFα and IL-6, providing a negative feedback mechanism [24]. It is possible that the up-regulation of TNFα and IL-6 inhibited IL-10 expression in the chicken embryo. 

MYD88 is a signal mediator between TLR4 and NFKB [25], but its expression was unaltered by LPS in chicken embryos although both the expression of both TLR4 and NFKB was upregulated. LPS stimulates TLR4-NFKB signaling occurs via several different TLR adapters including, MYD88, toll/IL-1R domain containing adaptor protein (TIRAP), TIR-domain-containing adapter-inducing IFN-β (TRIF), and TRIF-related adaptor molecules. One study indicated that chicken lack a functional LPS-specific TRAM-TRIF-signaling pathway [26]; another study demonstrated the gene expression of TLR4 and MYD88 was associated with the chicken breeds [27]. Therefore, in our inflammation model, the TLR4–NFKB signaling was probably related with the chicken breed. 

Extracellular matrix (ECM) is a complex macromolecule network which is composed of fibers, proteoglycans, glycoproteins and polysaccharides [28], and plays a critical pole in the cell morphology, functions, migration, proliferation, differentiation and survival through signaling pathways. Matrix metalloproteinases (MMPs), proteolytic enzymes of the ECM, are a zinc-dependent endopeptidase family mainly responsible for tissue remodeling in various physiological and pathological processes [29]. MMPs could degrade ECM proteins and glycoproteins, membrane receptors, cytokines, and growth factors. MMP3 was involved in initiating an early and lethal cytokine response to *Salmonella typhimurium* infection for mice [30]. Ulcerative colitis is a chronic inflammatory bowel disease in which the level of MMP3 and MMP9 is increased in human patients [31,32]. A previous study indicated intestinal MMP9 levels were increased in patients with inflammatory bowel disease [33] and in mice with colitis induced by dextran sodium sulfate [34]. Our results revealed the mRNA and protein levels of MMP3 and MMP9 induced by LPS were upregulated, whereas quercetin mitigated the MMP3 and MMP9 expression.

Autophagy refers to the process involving the decomposition of intracellular components via lysosomes. The intestinal autophagy is a complex system and plays a pivotal role in maintaining the homeostasis and integrity of the intestinal epithelium [35,36]. Autophagy impairment causes disruption of the intestinal epithelium, inappropriate immune response, and inflammation [37]. ATG5, LC3-1 and LC3-2 all participate in the formation of autophagosome. Paneth cells are the common cellular targets for ATG5 and ATG7 [38]. There are two type of ubiquitin-like binding reaction systems, namely ATG5-ATG7-ATG12 binding reaction systems and LC3-1/LC3-2 binding reaction systems. ATG5 is associated with the extension of the phagophore membrane in autophagic vesicles. ATG7 is involved in ATG12–ATG5 conjugation and LC3-1 lipidation in the autophagosomal membrane. LC3-1 is converted to LC3-2, which is essential in the elongation and closure of the autophagosome membrane for the initiation of autophagy. Nishino et al. found DSS-induced colitis was exacerbated in ATG5 deletion mice [39]. Our results show that the mRNA and relative protein expressions of ATG5, LC3-1, and LC3-2 were increased after LPS challenge, and downregulated by quercetin. It indicated quercetin could alleviate the duodenal autophagy induced by LPS. 

Multiple members of mechanistic target of rapamycin kinase (MTOR) pathways can regulate the initiation of autophagy, such as AMPK and MTOR complex 1 (MTORC1) [40]. AMPK is an upstream regulator of the MTOR pathway. LKB1 is involved in cellular metabolism, cellular polarity, and the DNA damage response. AMPK regulates the activation of autophagy by phosphorylation of ULK1. When energy deficiency, nutrient deprivation, and inflammation occurs, the energy-sensitive AMPK can be activated by LKB1 or autophagy-initiating kinase ULK1 triggering autophagy, while the MTORC1 is activated, inhibiting autophagy in nutrient-rich conditions [41]. Duan et al. found creatine nitrate supplementation strengthens energy status and delays the glycolysis of broiler muscle via inhibition of the mRNA expression of LKB1, AMPKA1, and AMPKA2 [42]. LKB1 triggers autophagy by directly phosphorylating AMPK. Quercetin inhibited LPS-induced autophagy probably by downregulation of LKB1.

LPS and/or inflammation can induce all three pathways of programmed cell death, apoptosis, pyroptosis and necroptosis. Thus, we also investigated the expression of genes representing these cell death pathways. LPS induced all three programmed cell death pathways, as shown by the upregulation of the pyroptosis-related genes CASP1 and CASP12, apoptosis-related genes Bcl-2, CASP3, and Fas, and necroptosis-related genes Drp1 and RIPK1. As for other LPS-induced changes, quercetin blocked the upregulation of all three programmed cell death pathways. Apoptosis, programmed cell death, is evolutionarily conserved across species, occurring regularly to eliminate excessive and potentially compromised cells to ensure cellular homeostasis. LPS could induce intestinal inflammation and apoptosis [43]. There were two classical signaling pathways to mediate apoptosis, namely the extrinsic or death receptor-dependent pathway and the intrinsic or mitochondria-dependent pathway. The extrinsic pathway is initiated by the binding of death ligands (Fas ligands, TNFa) to their receptors (Fas, TNFA1), caused by the recruitment of Fas-associated death domain adaptor molecule (FADD), FADD dimerizes and activates caspase 8, which cleaves and activates the procaspase 3 to CASP3, whereas the intrinsic pathway is activated by intracellular damage, oxidative stress and deprivation of growth factor. In the present study, the quercetin could reduce the Fas and TNFα expression after LPS induction. It suggested that the extrinsic apoptosis pathway was activated after LPS challenge, whereas quercetin could ameliorate the apoptotic functions in chicken embryos. Caspases, a family of cysteine aspartate proteases, are involved in apoptotic activities. There are three caspases groups: group I takes part in inflammation, and includes CASP 1, caspase 4, caspase 5, and CASP12; group II comprises initiator caspases including caspase 2, caspase 8, caspase 9; group III comprises executioner caspases including caspase 3, caspase 6, and caspase 7. CASP1 plays a vital pole in mediating tissue repair in intestinal homeostasis. CASP1 not only activates IL-1β by cleaving the pro-IL-1β, but also activates NFKB expression. CASP1-knockout mice are highly susceptible to diarrhea, inflammation initiation, and epithelial cell regeneration [44]. CASP3 plays a pivotal role in cell apoptosis, tissue differentiation and regeneration. NFKB activation in human inflammatory bowel disease is also correlated with the appearance of cleaved CASP3. We found the mRNA and protein levels of CASP1, IL-1β, and CASP3 increased after LPS challenge, and quercetin could decrease their expression in this study. Therefore, quercetin ameliorates LPS-induced apoptosis in intestinal inflammation in chicken embryos. 

Necroptosis, a caspase-independent and lytic programmed cell death displaying intermediate features between necrosis and apoptosis, is characterized by cell rupture and the release of cellular contents, and induces inflammation. The key actors are the receptor-interacting- serine/threonine protein-kinases 1 (RIPK1) and dynamin 1 like (Drp1). RIPK1 not only is an upstream kinase for Drp1 phosphorylation [45], but also is an upstream regulator of death receptors (TNFα) and pattern recognition receptors (TLRs) [46]. RIPK1 has been reported to maintain the homeostasis of cell survival, apoptosis and necroptosis. Evidence has indicated that RIPK1 depletion in intestinal epithelial cells spontaneously initiates severe intestinal inflammation associated with apoptosis in mice [47]. Activation of RIPK1 and RIPK3 involves a complex interplay between phosphorylation and ubiquitination to regulate necroptosis [48]. The RIPK1 maintains epithelial homeostasis by inhibiting caspase 8-depedent apoptosis and necroptosis [49]. Drp1 is an evolutionarily conserved member of the dynamin family and is responsible for mitochondrial division. Our results indicated the mRNA levels of RIPK1 and Drp1 increased after LPS induction, whereas quercetin alleviates duodenal necroptosis in chicken embryos. 

The intestinal mucosa provides a critical barrier of the body against unwanted foreign materials and its impairment may result in inflammation [50]. The intestinal mucosal barrier is maintained by epithelial tight junctions together with other structures. Tight junctions are composed of transmembrane proteins (claudins and occludin) and cytosolic scaffold proteins (tight junction proteins ZO-1 and cingulin). Thus, we determined whether quercetin protects tight junctions. While LPS markedly upregulated the tight junction proteins, claudin 1, occludin and ZO-1, quercetin completely blocked the upregulation of these genes and proteins. The increased expression of tight junction proteins in LPS-treated chicken embryos might be related to enhanced autophagy. Increased claudin 1 was observed in cancer cells from starvation-induced autophagy [51]. The mucus layer consists of mucins mostly produced by goblet cells, antimicrobial peptides secreted by Paneth cells, enterocytes and immune cells, and secretory IgA released by B lymphocytes. Mucin 2, a key mucus component of the mucus layer of the intestinal mucosa, was significantly down-regulated by LPS, but it was considerably upregulated by quercetin. Quercetin not only inhibited the effect of LPS on mucin 2, but also enhanced mucin 2, resulting in around an eight-fold increase in mRNA expression compared with the vehicle controls. The findings suggest quercetin may protect the intestinal barrier by maintaining epithelial tight junctions and stimulating mucin 2 synthesis and secretion. 

## 5. Conclusions

Quercetin is a main constituent of Chinese herbal medicines commonly used in the treatment of diarrhea. We aimed to avoid the interaction of environmental LPS and intestinal LPS from gut microorganisms. Therefore, the chicken embryos were selected for the present study. We found that quercetin may inhibit intestinal inflammation, which is a major cause of diarrhea, through multiple molecular pathways, including by inhibiting inflammatory cytokines, receptors, and transcription factors involved in inflammation, modulating autophagy, maintaining mucosa barrier integrity, and down-regulating programmed cell death (graphic abstract). The effects of quercetin on other organs will be revealed in our other manuscripts. Meanwhile, it warrants further study to testify the effects of kaempferol and beta-sitosterol in terms of their antidiarrheal activity.

## Figures and Tables

**Figure 1 animals-12-03524-f001:**
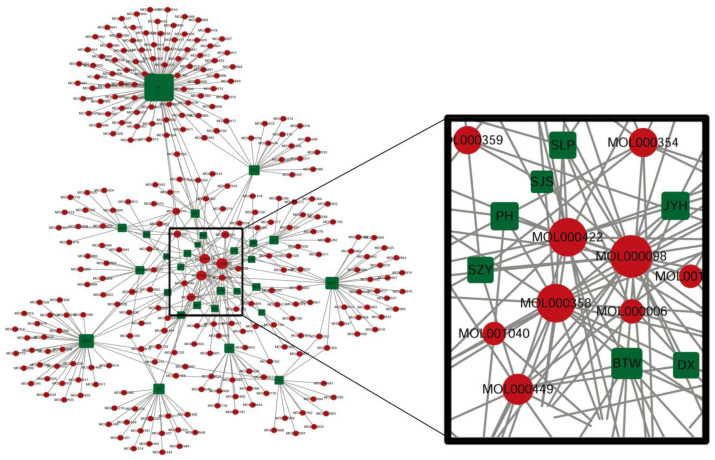
Bioactive ingredients of Chinese herbal medicines with antidiarrheal effects. Green square: Abbreviation of Chinese herbal medicines with antidiarrheal effect as shown in Table 1. Red circle: Bioactive ingredients. The selected Chinese herbal medicines were analyzed for bioactive ingredients on the Traditional Chinese Medicine Systems Pharmacology Database and Analysis platform (TCMSP; https://tcmspw.com/tcmsp.php (accessed from 10 November 2019 to 22 February 2021). Each selected Chinese herbal medicine contains 2 to 92 bioactive ingredients, yielding a total of 291 bioactive ingredients of the 28 Chinese herbal medicines. The 291 active ingredients were analyzed on Cytoscape (3.7.2 edition). Three active ingredients, quercetin, kaempferol, and beta-sitosterol, were selected based on the highest networking node degrees. Quercetin was presented in 23 Chinese herbal medicines in 28 ones (MOL000098: quercetin, node degree = 23), and kaempferol and beta-sitosterol in 19 Chinese herbal medicines in 28 ones (MOL000422: kaempferol, node degree = 19; MOL000358: beta-sitosterol, node degree = 19).

**Figure 2 animals-12-03524-f002:**
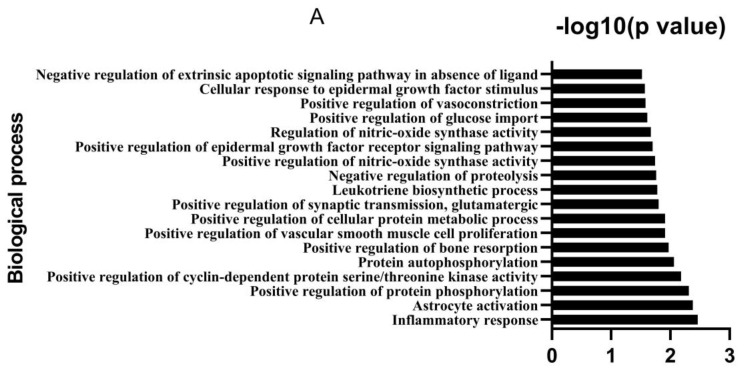
Gene ontology (GO) and the Kyoto encyclopedia of genes and genomes (KEGG) enrichment analysis of candidate gene targets associated with diarrhea. (**A**) Biological process; (**B**) Cellular components; (**C**) Molecular functions; (**D**) KEGG pathways. The target genes were subjected to gene ontology (GO) and Kyoto Encyclopedia of Genes and Genomes (KEGG) pathway enrichment analyses in the Database for Annotation, Visualization and Integrated Discovery (DAVID) to predict molecular biology functions, biology process, and cytology component, and signaling pathways (https://david-d.ncifcrf.gov/tools.jsp (accessed on 22 September 2020).

**Figure 3 animals-12-03524-f003:**
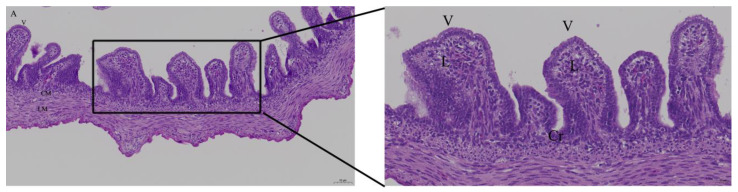
Histopathologic changes of the duodenum induced by LPS and treated with quercetin, with 40 nmol/egg in chicken embryos. Fifteen day-old embryonated eggs were inoculated with LPS (125 ng/egg) and quercetin (Q) at 40 nmol/egg by injection into the allantoic cavity. The duodenum was histologically examined on day 19 (4 days after treatment). Hematoxylin and eosin staining. (**A**) Control group; (**B**) There were inflammatory cell infiltration in the Peyer’s patch in LPS group (arrow); (**C**) Treatment group (125ng LPS/egg + 40 nmol quercetin/egg). The right photo was the rectangle of the left one magnified by 2 folds. Scale bar of left photo (200×): 50 μm. Scale bar of right photo (400×): 20 μm. V: villus; Cr: crypt; P: Peyer’s patch; L: lamina propria; CM: muscularis externa, inner circular; LM: muscularis externa, outer longitudinal.

**Figure 4 animals-12-03524-f004:**
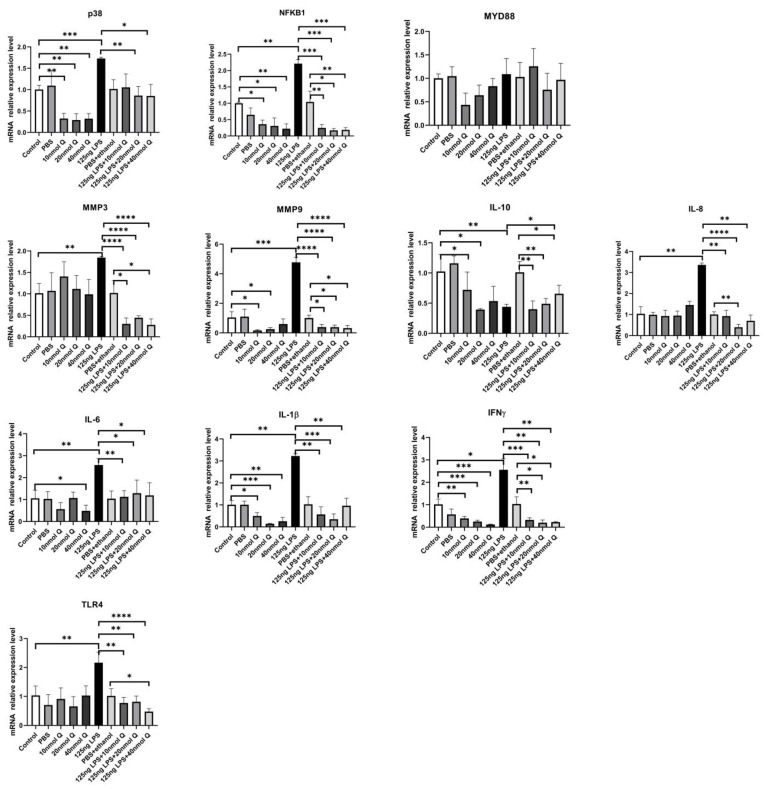
Quercetin attenuates LPS-induced duodenal inflammation. Fifteen day-old embryonated eggs were inoculated with LPS (125 ng/egg, dissolved in PBS) and quercetin (Q) at 10, 20, or 40 nmol/egg (dissolved in ethanol) or dosing vehicle (PBS, PBS + ethanol) by injection into the allantoic cavity. Duodenum tissue on day 19 was analyzed for mRNA expression of cytokines, enzymes or receptors by real-time PCR. Values and quantitative data are expressed as mean ± SD in each group. * *p* < 0.05; ** *p* < 0.01; *** *p* < 0.001; **** *p* < 0.0001.

**Figure 5 animals-12-03524-f005:**
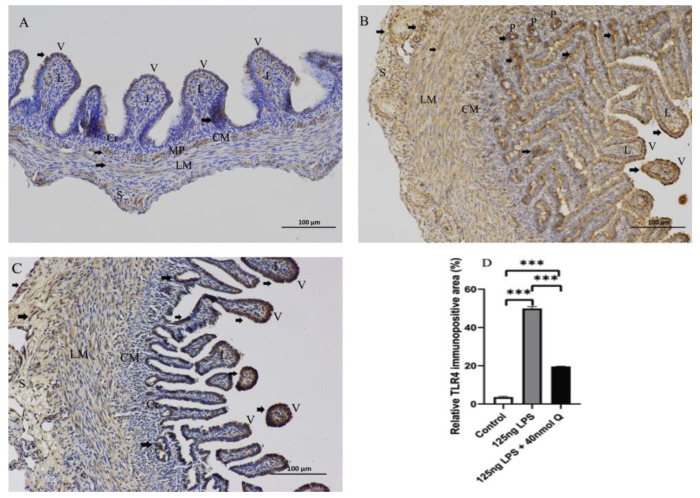
Immunohistochemical detection in HRP of TLR4 in the duodenum induced by LPS (125 ng/egg) and treated with 40nmol/egg quercetin in chicken embryos (×400). (**A**) Antibody for TLR4 revealed weak immunopositivity in the villi, myenteric plexus, tunica muscularis of control samples; (**B**) The immunopositivity in the villus, crypt, lamina propria, Peyer’s patch, tunica muscularis, and serosa of intestinal mucosa increased after LPS induction compared with that of control group; (**C**) The immunopositivity to TLR4 in treatment group (125 ng LPS/egg + 40 nmol Q/egg) significantly decreased when compared with that of LPS group; (**D**) Relative TLR4-immunopositive area. Immunopositivity to TLR4 (arrow, brown to yellow). Scale bar: 100 μm. V: villus; Cr: crypt; P: Peyer’s patch; L: lamina propria; CM: muscularis externa, inner circular; LM: muscularis externa, outer longitudinal; S: serosa; MP: myenteric plexus. Values and quantitative data are expressed as mean ± SD in each group. *** *p* < 0.001.

**Figure 6 animals-12-03524-f006:**
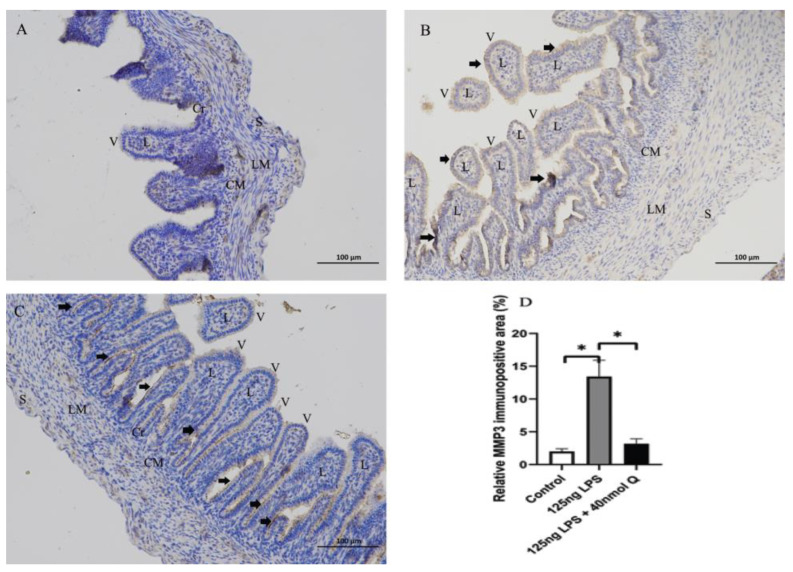
Immunohistochemical detection in HRP of MMP3 in the duodenum induced by LPS (125 ng/egg) and treated with 40nmol/egg quercetin in chicken embryos (×400). (**A**) Antibody for MMP3 revealed weak immunopositivity in the villi, crypts, tunica muscularis, and serosa of control samples; (**B**) The immunopositivity in the villus, crypt, and lamina propria increased after LPS induction compared with that of control group; (**C**) The immunopositivity to MMP3 in treatment group (125 ng LPS /egg + 40 nmol Q/egg) significantly decreased whhen compared with that of LPS group; (**D**) Relative MMP3-immunopositive area. Immunopositivity to MMP3 (arrow, brown to yellow). Scale bar: 100 μm. See the abbreviation for Figure 5. Values and quantitative data are expressed as mean ± SD in each group. * *p* < 0.05.

**Figure 7 animals-12-03524-f007:**
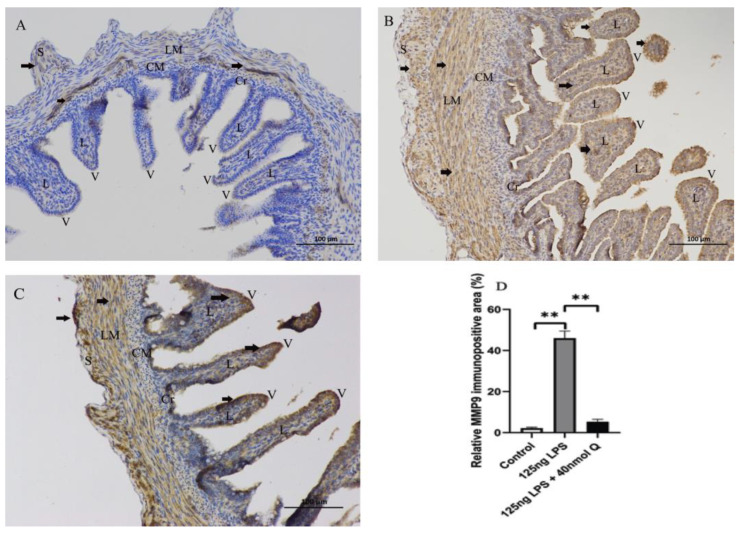
Immunohistochemical detection in HRP of MMP9 in the duodenum induced by LPS (125 ng/egg) and treated with 40nmol/egg quercetin in chicken embryos (×400). (**A**) Antibody for MMP9 revealed weak immunopositivity in the villi and tunica muscularis of intestinal mucosa of control samples; (**B**) The immunopositivity in the villus, crypt, lamina propria, and tunica muscularis of intestinal mucosa increased after LPS induction compared with that of control group; (**C**) The immunopositivity to MMP9 in treatment group (125 ng LPS /egg + 40 nmol Q/egg) significantly decreased when compared with that of LPS group; (**D**) Relative MMP9-immunopositive area. Immunopositivity to MMP9 (arrow, brown to yellow). Scale bar: 100 μm. See the abbreviation for Figure 5. Values and quantitative data are expressed as mean ± SD in each group. ** *p* < 0.01.

**Figure 8 animals-12-03524-f008:**
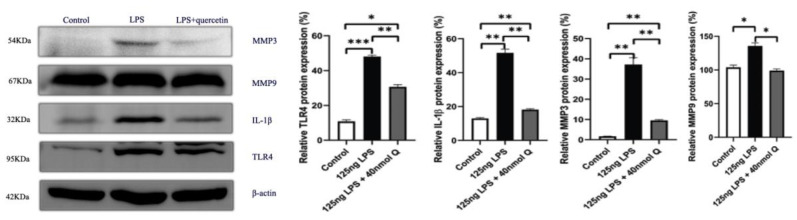
The protein expression level of TLR4, IL-1β, MMP3 and MMP9 after LPS induction in chicken embryos. Control, vehicle group; LPS, LPS group: 125 ng LPS /egg; LPS + quercetin: treatment group, (125 ng LPS + 40 nmol Q)/egg. Values and quantitative data are expressed as mean ± SD in each group. * *p* < 0.05; ** *p* < 0.01; *** *p* < 0.001.

**Figure 9 animals-12-03524-f009:**
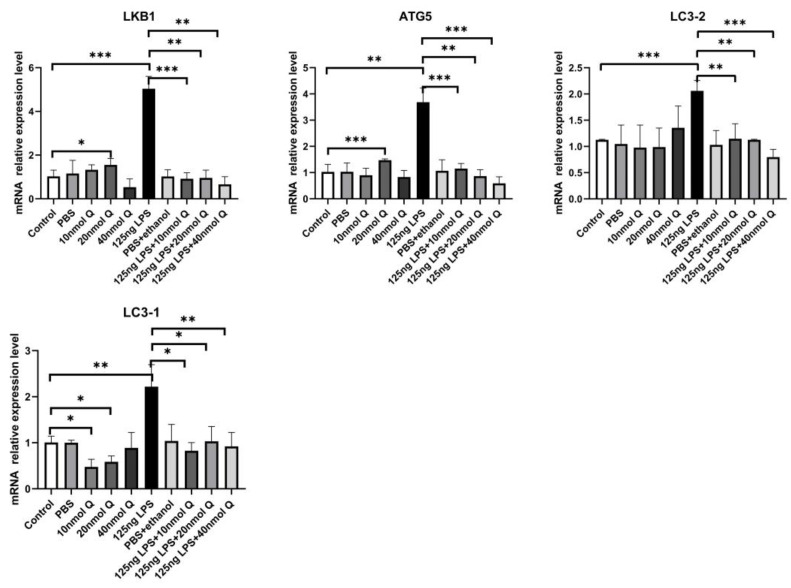
Quercetin ameliorates LPS-induced increase in expression of autophagy genes. See legends for Figure 4. Values and quantitative data are expressed as mean ± SD in each group. * *p* < 0.05; ** *p* < 0.01; *** *p* < 0.001.

**Figure 10 animals-12-03524-f010:**
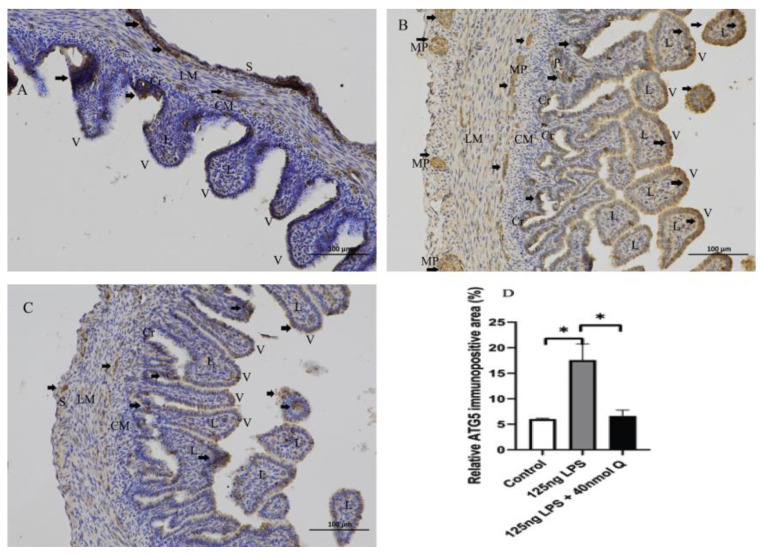
Immunohistochemical detection in HRP of ATG5 in duodenum induced by LPS (125 ng/egg) and treated with 40 nmol/egg quercetin in chicken embryos (×400). (**A**) Antibody for ATG5 revealed weak immunopositivity in the villi, crypts, and tunica muscularis of intestinal mucosa of control samples; (**B**) The immunopositivity in the villi, crypts, lamina propria, myenteric plexus, and tunica muscularis of intestinal mucosa increased after LPS induction compared with that of the control group; (**C**) The immunopositivity to ATG5 in the treatment group (125 ng LPS /egg + 40 nmol Q/egg) significantly decreased when compared with that of LPS group; (**D**) Relative ATG5-immunopositive area. Immunopositivity to ATG5 (arrow, brown to yellow). Scale bar: 100 μm. See the abbreviation for Figure 5. Values and quantitative data are expressed as mean ± SD in each group. * *p* < 0.05.

**Figure 11 animals-12-03524-f011:**
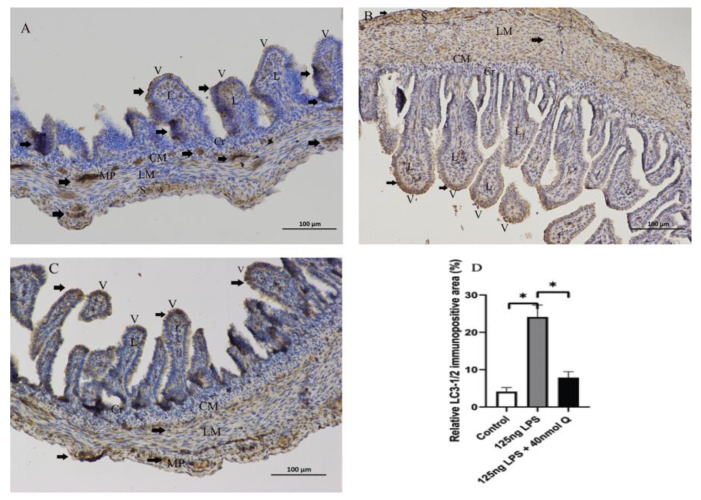
Immunohistochemical detection in HRP of LC3-1/2 in duodenum induced by LPS (125ng/egg) and treated with 40nmol/egg quercetin in chicken embryos (×400). (**A**) Antibody for LC3-1/2 revealed weak immunopositivity in the villi, crypts, and tunica muscularis of intestinal mucosa of control samples; (**B**) The immunopositivity in the villus, crypt, lamina propria, and tunica muscularis increased after LPS induction compared with that of the control group; (**C**) The immunopositivity to LC3-1/2 I then treatment group (125 ng LPS /egg + 40 nmol Q/egg) significantly decreased which compared with that of the LPS group; (**D**) Relative LC3-1/2-immunopositive area. Immunopositivity to LC3-1/2 (arrow, brown to yellow). Scale bar: 100 μm. See the abbreviation for Figure 5. Values and quantitative data are expressed as mean ± SD in each group. * *p* < 0.05.

**Figure 12 animals-12-03524-f012:**
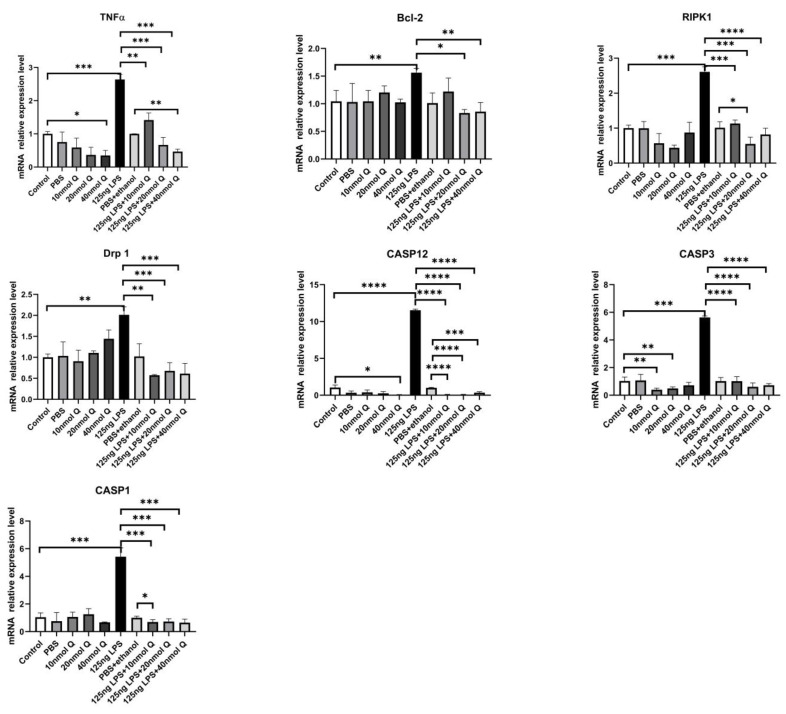
Quercetin inhibits LPS-induced programmed cell death. See legends for Figure 4. Values and quantitative data are expressed as mean ± SD in each group. * *p* < 0.05; ** *p* < 0.01; *** *p* < 0.001; **** *p* < 0.0001.

**Figure 13 animals-12-03524-f013:**
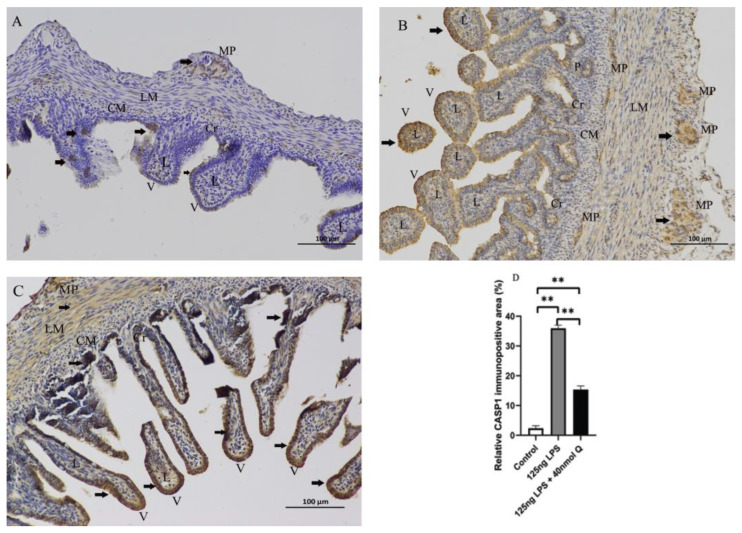
Immunohistochemical detection in HRP of CASP1 in duodenum induced by LPS (125 ng/egg) and treated with 40nmol/egg quercetin in chicken embryos (×400). (**A**) Antibody for CASP1 revealed weak immunopositivity in the villi, crypts, and tunica muscularis of control samples; (**B**) The immunopositivity in the villi, crypts, lamina propria, myenteric plexus, Peyer’s patch, and tunica muscularis increased after LPS induction compared with that of the control group; (**C**) The immunopositivity to CASP1 in the treatment group (125 ng LPS /egg + 40 nmol Q/egg) significantly decreased when compared with that of the LPS group; (**D**) Relative CASP1-immunopositive area. Immunopositivity to CASP1 (arrow, brown to yellow). Scale bar: 100 μm. See the abbreviation for Figure 5. Values and quantitative data are expressed as mean ± SD in each group. ** *p* < 0.01.

**Figure 14 animals-12-03524-f014:**
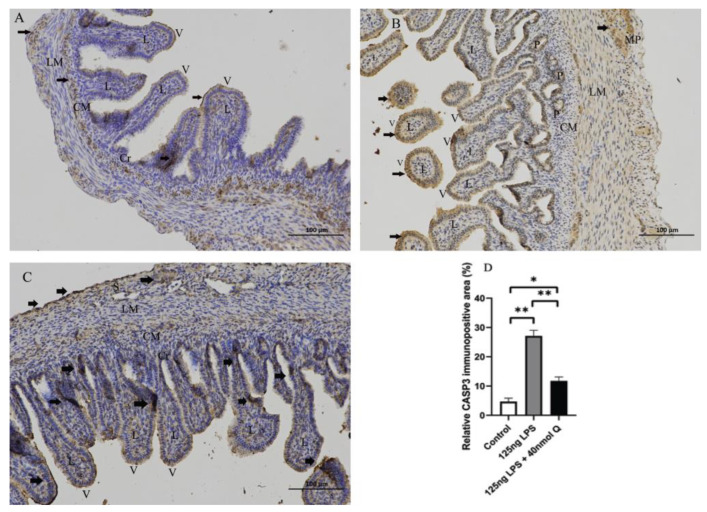
Immunohistochemical detection in HRP of CASP3 in duodenum induced by LPS (125 ng/egg) and treated with 40nmol/egg quercetin in chicken embryos (×400). (**A**) Antibody for CASP3 revealed weak immunopositivity in the villi, crypts, lamina propria, and tunica muscularis of intestinal mucosa of control samples; (**B**) The immunopositivity in the villus, crypt, lamina propria, myenteric plexus, Peyer’s patch, and tunica muscularis increased after LPS induction compared with that of the control group; (**C**) The immunopositivity to CASP3 in treatment group (125 ng LPS /egg + 40 nmol Q/egg) significantly decreased when compared with that of LPS group; (**D**) Relative CASP3-immunopositive area. Immunopositivity to CASP3 (arrow, brown to yellow). Scale bar: 100 μm. See the abbreviation for Figure 5. Values and quantitative data are expressed as mean ± SD in each group. * *p* < 0.05; ** *p* < 0.01.

**Figure 15 animals-12-03524-f015:**
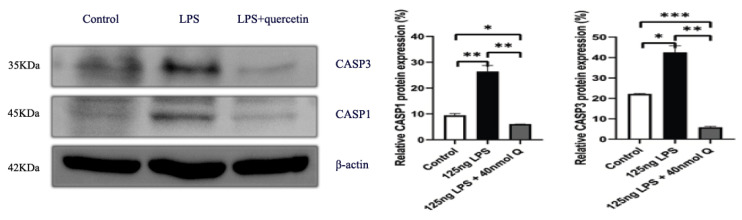
The protein expression level of CASP1 and CASP3 after LPS induction in chicken embryos. Control, vehicle group; LPS, LPS group: 125ng LPS /egg; LPS + quercetin: treatment group, (125 ng LPS + 40 nmol Q)/egg. Values and quantitative data are expressed as mean ± SD in each group. * *p* < 0.05; ** *p* < 0.01; *** *p* < 0.001.

**Figure 16 animals-12-03524-f016:**
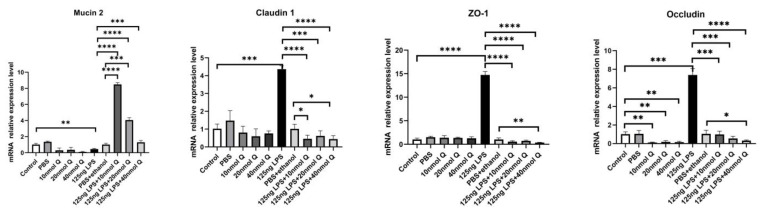
Quercetin ameliorates LPS-induced impairment of intestinal mucosal barrier functions. See legends for Figure 4. Values and quantitative data are expressed as mean ± SD in each group. * *p* < 0.05; ** *p* < 0.01; *** *p* < 0.001; **** *p* < 0.0001.

**Figure 17 animals-12-03524-f017:**
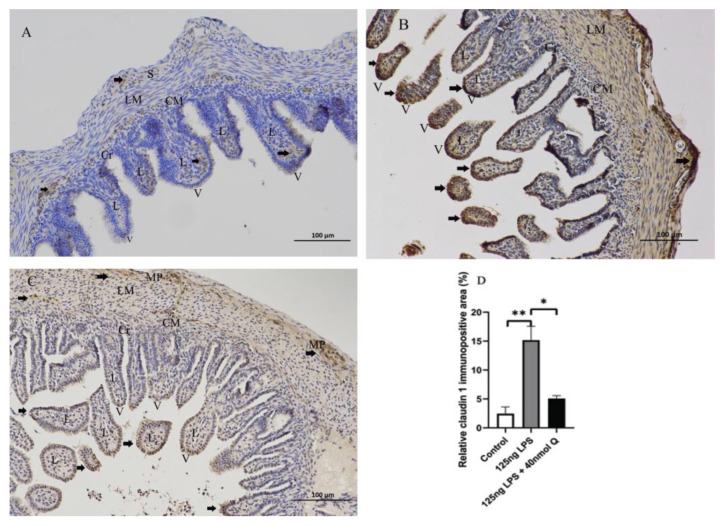
Immunohistochemical detection in HRP of claudin 1 in duodenum induced by LPS (125 ng/egg) and treated with 40 nmol/egg quercetin in chicken embryos (×400). (**A**) Antibody for claudin 1 revealed weak immunopositivity in the lamina propria, tunica muscularis, and serosa of intestinal mucosa of control samples, and no immunoreactivity in the intestinal villi and crypts; (**B**) The immunopositivity in the villus, crypt, lamina propria, and tunica muscularis of intestinal mucosa increased after LPS induction compared with that of the control group; (**C**) The immunopositivity to claudin 1 in treatment group (125 ng LPS /egg + 40 nmol Q/egg) significantly decreased when compared with that of the LPS group; (**D**) Relative claudin 1-immunopositive area. Immunopositivity to claudin 1 (arrow, brown to yellow). Scale bar: 100 μm. See the abbreviation for Figure 5. Values and quantitative data are expressed as mean ± SD in each group. * *p* < 0.05; ** *p* < 0.01.

**Figure 18 animals-12-03524-f018:**
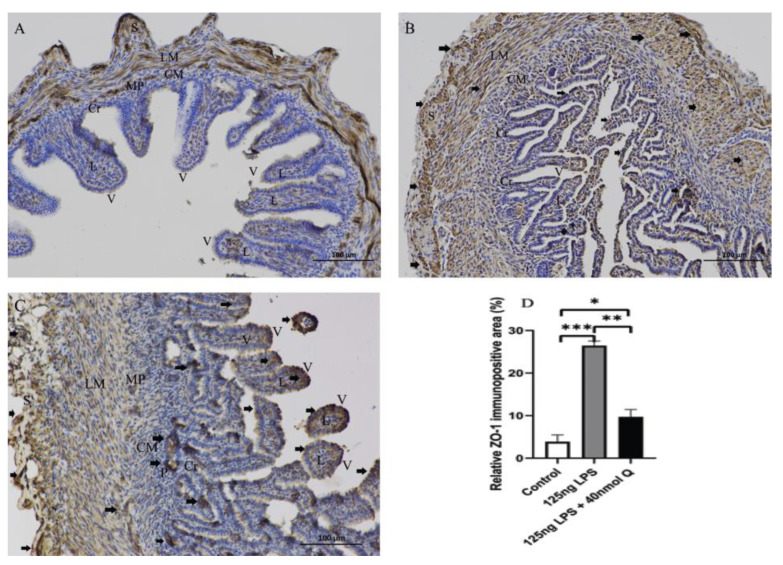
Immunohistochemical detection in HRP of ZO-1 in duodenum induced by LPS (125 ng/egg) and treated with 40nmol/egg quercetin in chicken embryos (×400). (**A**) Antibody for ZO-1 revealed immunopositivity in the lamina propria and tunica muscularis of intestinal mucosa of control samples, and weak immunoreactivity in the intestinal villi and crypts; (**B**) The immunopositivity in the villus, crypt, lamina propria, and tunica muscularis of intestinal mucosa increased after LPS induction compared with that of the control group; (**C**) The immunopositivity to ZO-1 in treatment group (125 ng LPS /egg + 40 nmol Q/egg) significantly decreased when compared with that of control group; (**D**) Relative ZO-1-immunopositive area. Immunopositivity to ZO-1 (arrow, brown to yellow). Scale bar: 100 μm. See the abbreviation for Figure 5. Values and quantitative data are expressed as mean ± SD in each group. * *p* < 0.05; ** *p* < 0.01; *** *p* < 0.001.

**Figure 19 animals-12-03524-f019:**
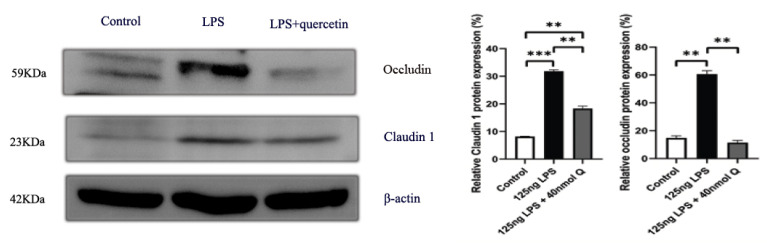
The protein expression level of claudin 1 and occludin after LPS induction in chicken embryos. Control, vehicle group; LPS, LPS group: 125 ng LPS /egg; LPS + quercetin: treatment group, (125 ng LPS + 40 nmol Q)/egg. Values and quantitative data are expressed as mean ± SD in each group. ** *p* < 0.01; *** *p* < 0.001.

**Table 1 animals-12-03524-t001:** Chinese herbal medicines with antidiarrheal effects.

Botanic Name	English Common Name	Name in Chinese Pinyin	Abbreviation
*Folium Artemisiae argyi*	Moxa	Aiye	AIYE
*Paeoniae Radix Alba*	White peony root	Baishao	BS
*Pulsatilliae Radix*	Chinese *Pulsatilla* root	Baitouweng	BTW
*Platycladi cacumen*	*Platycladus orientalis* leaves	Cebaiye	CBY
*Radix Bupleuri*	Chinese thorowax root	Chaihu	CH
*Ailanthi Cortex*	Bark of ailanthus	Chunpi	CP
*Radix Sanguisorbae* officinalis L.	Garden burnet root	Diyu	DIYU
*Caryophylli flos*, *Syzygium aromaticum*	Clove	Dingxiang	DX
*Glycyrrhizae radix et Rhizoma*	Licorice	Gancao	GC
*Alpiniae officinarum Rhizome*	Galangal	Gaoliangjiang	GLJ
*Zanthoxylum bungeanum* Maxim. *Pericarpium*	Pepper pericarp	Huajiao	HJ
*Coptidis Rhizoma*	Coptis root	Huanglian	HL
*Hedysarum Multijugum Maxim.*, *Astragalus membranaceus*	Astragali radix	Huangqi	HQ
*Scutellariae Radix*	Baical skullcap root	Huangqin	HQIN
*Lonicerae Japonicae Flos*	Honeysuckle flower	Jinyinhua	JYH
*Rosae Laevigatae Fructus*	Cherokee rose fruit	Jinyingzi	JYZ
*Portulacae Herba*	Purslane	Machixian	MCX
*Chaenomeles sinensis (Thouin) Koehne*	Pawpaw	Mugua	MG
*Aucklandiae Radix*	Costustoot	Muxiang	MX
*Equiseti Hiemalis Herba*	Scouring rush herb	Muzei	MZ
*Pollen Typhae*	Cattail pollen	Puhuang	PH
*Panax ginseng C. A. Mey.*	Ginseng	Renshen	RS
*Herba Taxilli*	Chinese taxillus twig	Sangjisheng	SJS
*Granati Pericarpium*	Pomegranate bark	Shiliupi	SLP
*Crataegi folium*	Hawthorn leaf	Shanzhaye	SZY
*Mume Fructus*	Dark plum fruit	Wumei	WM
*Evodiae fructus*	Evodia fruit	Wuzhuyu	WZY
*Leonuri Herba*	Motherwort	Yimucao	YMC

**Table 2 animals-12-03524-t002:** Primers used in real-time quantitative polymerase chain reaction.

Genes Name	Primer Sequence (5′-3′)	Gene Bank ID
**ATG5**	F:CCCATCCCTGGTCCGTAAC; R:CGGCGGCGTATACGAAGTA	NM_001006409
**Bcl-2**	F: TGGCTGCTTTACTCTTGGGG; R:TATCTCGCGGTTGTCGTAGC	NM_205339
**CASP1**	F:CACTTCCACTTCGGATGGCT; R:CCACGAGACAGTATCAGGCG	XM_015295935
**CASP3**	F:ACCGAGATACCGGACTGTCA; R:GCCATGGCTTAGCAACACAC	NM_204725
**CASP12**	F:AATAGTGGGCATCTGGGTCA; R:CGGTGTGATTTAGACCCGTAAGAC	[7]
**Claudin 1**	F:CTGGGTCTGGTTGGTGTGTT; R:CGAGCCACTCTGTTGCCATA	NM_001013611
**Drp1**	F: GGCAGTCACAGCAGCTAACA; R:GCATCCATGAGATCCAGCTT	NM_001079722
**Fas**	F:GTCAGTGCTGCACGAAATGT; R:AACCTCCAAACCGAGTGCTT	NM_001199487
**GAPDH**	F: GAGAAACCAGCCAAGTATGATG; R: CACAGGAGACAACCTGGTCC	NM_204305
**IFNγ**	F:CTGACAAGTCAAAGCCGCAC; R:CTTCACGCCATCAGGAAGGT	NM_205149
**IL-10**	F:TGCGAGAAGAGGAGCAAAGC; R:AACTCCCCCATGGCTTTGTAG	AJ621254
**IL-1β**	F:GCTCAACATTGCGCTGTACC; R:AGGCGGTAGAAGATGAAGCG	FJ537850
**IL-6**	F:ACGAGGAGAAATGCCTGACG; R:CTTCAGATTGGCGAGGAGGG	NM_204628
**IL-8**	F:TGCCAGTGCATTAGCACTCA; R:TTGGCGTCAGCTTCACATCT	HM179639
**LC3-1**	F:GCATCCAAACAAAATCCCAGTC; R:AAGCCATCCTCATCCTTCTCCT	XM_040688401
**LC3-2**	F:CTTCTTCCTCCTGGTGAACG; R:GCACTCCGAAAGTCTCCTGA	NM_001031461
**LKB1**	F:AGCAGAGGCATTGCATCCAT; R:CCTGCGGACCAGATGTCTAC	NM_001045833
**MMP9**	F:ACACAGACTCTATGCTGCCTG; R:GAGAGTAGGGCGGGGAAAAT	NM_204667
**MMP3**	F:ATCAGGCTCTACAGTGGTG; R:ATGGGATACATCAAGGCAC	XM_025152201
**Mucin 2**	F:TCAGATCAGATGGCAGTGTGTC; R:AATCTGCAGCTGAAGCCCAAA	JX284122
**MYD88**	F:TTAGTCTTTCCCCAGGGGCT; R:GCCAGTCTTGTCCAGAACCA	NM_001030962
**NFKB1**	F:TCAACGCAGGACCTAAAGACAT; R:GCAGATAGCCAAGTTCAGGATG	NM_001396396
**Occludin**	F:TACATCATGGGCGTCAACCC; R:CCAGATCTTACTGCGCGTCT	NM_205128
**p38**	F:GGTCGGTGAGCTGGTAAAGG; R:CGCTTTCAGCTTCTGTCGGA	XM_015296032
**RIPK1**	F:GATCCATTTGCGAAGCTGCC; R:CTTAGGCTAATGGCGCTGGT	NM_204402
**TLR4**	F:GGCTCAACCTCACGTTGGTA; R:AGTCCGTTCTGAAATGCCGT	KP410249
**ZO-1**	F:TATGAAGATCGTGCGCCTCC; R:GAGGTCTGCCATCGTAGCTC	XM_015278977
**TNFα**	F:CCCATCCCTGGTCCGTAAC; R:CGGCGGCGTATACGAAGTA	MF000729

Abbreviation: ATG5: autophagy-related gene 5; Bcl-2: B cell CLL/lymphoma 2; CASP1: caspase 1; CASP3: caspase 3; CASP12: caspase 12; Drp1, dynamin 1 like; GAPDH: Glyceraldehyde-3-phosphate dehydrogenase; Fas: Fas cell surface death receptor; IFNγ: interferon gamma; IL-10: interleukin-10; IL-8: interleukin-8; IL-1β: interleukin-1β; IL-6: interleukin-6; LC3-1: microtubule associated protein 1 light chain 3 alpha; LC3-2: microtubule-associated protein 1 light chain 3 beta; LKB1, serine/threonine kinase 11; MMP3: matrix metallopeptidase 3; MMP9: matrix metallopeptidase 9; MYD88: myeloid differentiation primary response 88;NFKB1: nuclear factor kappa B subunit 1; RIPK1: receptor interacting serine/threonine kinase 1; TLR4: toll like receptor 4;ZO-1: Tight junction protein 1; TNFA: tumor necrosis factor alpha.

**Table 3 animals-12-03524-t003:** Target genes of quercetin, kaempferol and beta-sitosterol from analyses on SwissTargetPrediction, OMIM and GeneCard.

Gene ID	Gene Name Abbreviation	Gene Name
5243	ABCB1	ATP Binding Cassette Subfamily B Member 1)
135	ADORA2A	Adenosine A2a Receptor
196	AHR	Aryl Hydrocarbon Receptor
207	AKT1	AKT Serine/Threonine Kinase 1
240	ALOX5	Arachidonate 5-Lipoxygenase
3577	CXCR1	C-X-C Motif Chemokine Receptor 1
1956	EGFR	Epidermal Growth Factor Receptor
2100	ESR2	Estrogen Receptor 2
4314	MMP3	Matrix Metallopeptidase 3
4318	MMP9	Matrix Metallopeptidase 9
4350	MPG	N-Methylpurine DNA Glycosylase
6850	SYK	Spleen Associated Tyrosine Kinase
7015	TERT	Telomerase Reverse Transcriptase

## Data Availability

Not applicable.

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
