# Peer review of "Quercetin Ameliorates Lipopolysaccharide-Induced Duodenal Inflammation through Modulating Autophagy, Programmed Cell Death and Intestinal Mucosal Barrier Function in Chicken Embryos"

_animals, 2022, doi:10.3390/ani12243524_

Round 1

Reviewer 1 Report

Its very good manuscript and very well organised with extensive modern analytical technuices and findings. Atuhors may add references on quercetin on broilers by the studies of Prof Simitzis, Prof Mountzouris and Prof Goliomytis that provided interesting in vivo findings with quercetin on oxidation, gene expression and overal antioxidant quality.

Authors must further clarify the quercetin test in comparison with other chinese herbal remedies.

Author Response

Reviewer 1

It’s very good manuscript and very well organised with extensive modern analytical technuices and findings. Atuhors may add references on quercetin on broilers by the studies of Prof Simitzis, Prof Mountzouris and Prof Goliomytis that provided interesting in vivo findings with quercetin on oxidation, gene expression and overal antioxidant quality.

Response:

These references had been added in introduction and discussion.

In introduction

Diarrhea is a global health problem for centuries. Since 1990, diarrhea has been ranked among the top ten causes of morbidity and mortality in all age groups [1]. Most diarrheal cases are caused by bacteria, parasites, or viruses, with gram-negative bacteria, such as Escherichia coli, Shigella, Salmonella [2], Campylobacter, and vibrio species among the most common pathogens.

In discussion

Quercetin is a common flavonoid of extracts of vegetables, fruits, and traditional Chinese herbs, it has antioxidant, anti-inflammatory, hepatoprotective, and anti-cancer effects [8]. It reported the quercetin dietary supplementation had antioxidation effect on meat of broiler chicken [9]. Another study indicated quercetin supplementation altered the elemental profiles in laying hens[10].

Authors must further clarify the quercetin test in comparison with other chinese herbal remedies.

Response:

We found quercetin, kaempferol, and beta-sitosterol in 28 Chinese herbs had antidiarrheal effect through network pharmacology. Evidences indicated quercetin and kaempferol had bacteriostatic effect on gram-positive and gram-negative bacteria, such as Streptococcus mutans [11], Salmonella enterica serotype Typhimurium, and Escherichia coli [12].

We will study the antidiarrheal mechanism of kaempferol and beta-sitosterol in the future, including the inflammation, autophagy, programed cell deaths.

Reviewer 2 Report

Dear Authors 

Your experimental work has been well described. Please improve the identification of several figure . Letters inside  the photo some are not well visualize

Conclusion is based on the data obtained . However due the experiment was conducted under embryo process, It must be important to  state that specific point in the conclusion section. 

Author Response

Reviewer 2

Your experimental work has been well described. Please improve the identification of several figure . Letters inside  the photo some are not well visualize.

Response:

The letters inside figures had been magnified, and the photos were visualized.

Conclusion is based on the data obtained. However due the experiment was conducted under embryo process, It must be important to state that specific point in the conclusion section. 

Response:

In the Methods:

Because the chick genome demonstrates remarkable evolutionary conservation with mammals, the expression patterns of several gene and protein are well-conserved between chick and mouse embryos. In addition, injection into the allantoic cavity of chicken embryos was ideal method to avoid the interaction of environmental LPS and intestinal LPS from gut microorganisms. Therefore, the chicken embryos were selected for present study.

Conclusions:

Quercetin is a main constituent of Chinese herbal medicines commonly used in the treatment of diarrhea. To avoid the interaction of environmental LPS and intestinal LPS from gut microorganisms. Therefore, the chicken embryos were selected for present study.Our findings that quercetin may inhibit intestinal inflammation, which is a major cause of diarrhea, through multiple molecular pathways by inhibiting inflammatory cytokines, receptors, and transcription factors involved in inflammation, modulating autophagy, maintaining mucosa barrier integrity, and down-regulating programmed cell deaths (Figure 20, graphic abstract). However, it warrants further study to testify the effects of kaempferol and beta-sitosterol.

Reviewer 3 Report

The manuscript is interesting, but many aspects must be considered before publication.

To mention the most important, extensive editing of the English language and style is required. In addition, authors should review the instructions for authors since the length of the abstract is greater than that established.

In the introduction, the final paragraph includes results obtained from the investigation, when it usually ends with the objective of the study.

There are parts of the methodology that must be rewritten since it is not understood what the authors are implying. Likewise, L. 238 talks about liver samples and these samples were never mentioned or results are shown, please review.

Don't you think that the size of the groups is quite small and inadequate to be able to reach reliable conclusions?

The discussion is very long and in some parts repetitive. The authors must synthesize it in order to show the most important elements of the investigation.

The conclusion must be rewritten since it is not understood

Author Response

Response to reviewers 3

The manuscript is interesting, but many aspects must be considered before publication.

To mention the most important, extensive editing of the English language and style is required. In addition, authors should review the instructions for authors since the length of the abstract is greater than that established.

Response:

This manuscript had been revised by one American, and one Australian and one Italian. Now it had been revised again, please see the revised manuscript.

In the introduction, the final paragraph includes results obtained from the investigation, when it usually ends with the objective of the study.

Response:

The final paragraph had been moved into methods part:

We found quercetin, kaempferol, and beta-sitosterol in 28 Chinese herbal medicines with antidiarrheal activity. The biological processes of diarrhea were involved in inflammatory response. TNF signaling pathway plays an important pole in diarrhea which contains apoptosis, autophagy, necroptosis and inflammatory cytokines expression. To verify in silico analysis findings, LPS from salmonella enterica serotype typhimurium was selected to induce duodenal inflammation in embryonated chicken eggs to simulate diarrhea, and quercetin, as a representative active ingredient, was administered to demonstrate antidiarrheal effects via modulating autophagy, apoptosis, necroptosis, and intestinal mucosal barrier functions.

There are parts of the methodology that must be rewritten since it is not understood what the authors are implying. Likewise, L. 238 talks about liver samples and these samples were never mentioned or results are shown, please review.

Response:

The line 238 had been revised, and deleted the word “liver”.

Don't you think that the size of the groups is quite small and inadequate to be able to reach reliable conclusions?

Response:

The fertilized eggs were individually weighed before hatching and divided into 10 groups, each group consisting of 4 replicates with 3 eggs per replicate.

Therefore, the duodenum of 12 fertilized eggs were collected for gene and protein expression (n=12), it appeared had no such question on the small size of the groups.

The discussion is very long and in some parts repetitive. The authors must synthesize it in order to show the most important elements of the investigation.

Response:

Because this manuscript found quercetin, kaempferol, and beta-sitosterol in 28 Chinese herbal medicines with antidiarrheal activity by network pharmacology. It had mainly discussed the gene and proteins expressions of quercetin ameliorates lipopolysaccharide induced duodenal inflammation through modulating autophagy, programmed cell deaths and intestinal mucosal barrier function in chicken embryos, which including 28 genes and 11 proteins in this manuscript, therefore, it need to discuss these changes after LPS induction with quercetin treatment, it appeared long in the discussion.

The conclusion must be rewritten since it is not understood.

Response:

The conclusion had been revised, such as follows:

 Quercetin is a main constituent of Chinese herbal medicines commonly used in the treatment of diarrhea. To avoid the interaction of environmental LPS and intestinal LPS from gut microorganisms. Therefore, the chicken embryos were selected for present study. Our findings that quercetin may inhibit intestinal inflammation, which is a major cause of diarrhea, through multiple molecular pathways by inhibiting inflammatory cytokines, receptors, and transcription factors involved in inflammation, modulating autophagy, maintaining mucosa barrier integrity, and down-regulating programmed cell deaths (Figure 20, graphic abstract). The effects of quercetin on other organs will be revealed in our other manuscripts. Meanwhile, it warrants further study to testify the effects of kaempferol and beta-sitosterol antidiarrheal activity.

Round 2

Reviewer 2 Report

Dear Authors

I just read your reactions and it seems the conclusion right now improve the outcome . 

Reviewer 3 Report

The authors partially considered the suggestions made in the first round.